# The Influence of Early Onset Preeclampsia on Perinatal Red Blood Cell Characteristics of Neonates

**DOI:** 10.3390/ijms24108496

**Published:** 2023-05-09

**Authors:** Barbara Sandor, Beata Csiszar, Gergely Galos, Simone Funke, Dora Kinga Kevey, Matyas Meggyes, Laszlo Szereday, Kalman Toth

**Affiliations:** 11st Department of Medicine, Medical School, University of Pécs, 7624 Pécs, Hungary; sandor.barbara@pte.hu (B.S.); galos.gergely@pte.hu (G.G.); 2Department of Anaesthesiology and Intensive Therapy, Medical School, University of Pécs, 7624 Pécs, Hungary; csiszar.beata@pte.hu; 3Department of Obstetrics and Gynaecology, Medical School, University of Pécs, 7624 Pécs, Hungary; funke.simone@pte.hu (S.F.); keveydorka@gmail.com (D.K.K.); 4Department of Medical Microbiology and Immunology, Medical School, University of Pécs, 7624 Pécs, Hungary; meggyes.matyas@pte.hu (M.M.); szereday.laszlo@pte.hu (L.S.)

**Keywords:** preeclampsia, red blood cell properties, neonatal adaptation, intrauterine growth restriction

## Abstract

Preeclampsia is the leading cause of complicated neonatal adaptation. The present investigation aimed to study the hemorheological factors during the early perinatal period (cord blood, 24 and 72 h after delivery) in newborns of early-onset preeclamptic mothers (*n* = 13) and healthy neonates (*n* = 17). Hematocrit, plasma, and whole blood viscosity (WBV), red blood cell (RBC) aggregation, and deformability were investigated. There were no significant differences in hematocrit. WBV was significantly lower in preterm neonates at birth than in the term 24 and 72 h samples. Plasma viscosity was significantly lower in preterm neonates’ cord blood than in healthy controls. RBC aggregation parameters were significantly lower in preterm newborns’ cord blood than in term neonates’ cord blood 24 and 72 h samples. RBC elongation indices were significantly lower in the term group than in preterm neonates 72 h’ sample at the high and middle shear stress range. Changes in the hemorheological parameters, especially RBC aggregation properties, refer to better microcirculation of preterm neonates at birth, which could be an adaptation mechanism to the impaired uteroplacental microcirculation in preeclampsia.

## 1. Introduction

Preeclampsia seriously contributes to the incidence of preterm birth and affects ca. 5–8% of gravidities [1,2]. It is responsible for 14% of all maternal deaths. According to the World Health Organisation (WHO), early-onset preeclampsia means maternal hypertension of previously healthy mothers diagnosed before the 20th gestational week [3,4]. The disease shows itself in proteinuria, target organ failure (e.g., renal or hepatic disorders, hematological disturbances, neurological symptoms such as eclamptic seizure) and uteroplacental failure resulting in intrauterine growth retardation (IUGR) of the neonate [5]. The leading cause of IUGR is uteroplacental hypoperfusion due to endothelial dysfunction of the uterine arteries, the cause of which is not fully uncovered, but oxidative stress has an important role in it [1]. Secondary to hypoperfusion, fetal hypoxia and undersupply of nutrients end in intrauterine growth retardation and preterm birth. IUGR has a pivotal role in the complicated adaption of newborns and also contributes to their higher cardiovascular risk in adulthood [6,7].

Previous studies have already investigated maternal blood rheology in preeclampsia. Only a few focused on fetal rheological parameters, although the rheological properties of blood are particularly important for maintaining blood flow and microcirculation in newborns, especially due to their low mean blood pressure [8,9]. The abnormal maternal rheological changes due to preeclampsia might exert harmful effects on the microcirculation of the placenta leading to damaged tissue oxygenation in the fetal organs such as the kidney, brain, or intestines, resulting in disturbed adaptation to extrauterine life [10]. Most rheological studies investigated only umbilical cord blood and gained conflicting information. Hence, an investigation of perinatal hemorheology in preterm infants of preeclamptic mothers could be a gap filler.

The aim of our study was to evaluate hemorheological parameters in newborns of mothers with early-onset preeclampsia by investigating red blood cell (RBC) properties. With the examination of healthy neonates, we intended to obtain a complete picture of the normal rheological alterations in the fetus and the normal perinatal period. Our study focused on obtaining novel information about the early neonatal period concerning the hemorheological changes in neonates with maternal preeclampsia.

## 2. Results

### 2.1. Population Characteristics

The statistical analyses of the descriptive data revealed the expected significant differences between the preterm and healthy neonates. As expected, preterm neonates were born with significantly lower gestational age, with significantly lower weight and length than healthy controls. Moreover, 1 min and 5 min Apgar scores were also significantly lower in preterm neonates than in newborns at term. In accordance with these findings, the weight percentile is significantly lower; therefore, IUGR is significantly more frequent in preterm newborns, which led to considerably longer hospitalisation periods in this patient group compared to healthy babies (Table 1). There was no significant difference in the usage of delayed cord clamping (delayed by 30–60 s). During the hospitalization period, we recorded 1 complication in the term group (collarbone fracture), but a total of 10 in the preterm group (2 ROP cases in stadium >II, 2 BPD, 3 IVH, 2 cardiac problems, 1 meconium ileus). Upon 1 April 2023, we have observed no deaths, but we have registered one hospitalization on recurrent airway infection in the term group. Three preterm infants were hospitalized (each of them more than twice) because of severe airway infection and gastrointestinal problems (Table 1).

### 2.2. Hemorheological Findings

The investigation revealed no significant differences within and between preterm and term neonates’ hematocrit values at any time (Table 2).

The investigation of whole blood viscosity (WBV) showed no statistically significant differences in the preterm group. There were no significant alterations in the term group, but there was a noticeable increase in the postnatal period. In contrast, WBV remained almost the same perinatally among the preterm newborns (Table 3 and Figure 1). Our study presented significantly lower values in the preterm group at birth than in the term group 24 (*p* = 0.001) and 72 h (*p* = 0.023) after delivery. Plasma viscosity (PV) measured at birth showed significantly lower values in preterm than in term neonates (Table 3).

### 2.3. RBC Aggregation

The measured RBC aggregation parameters showed univocally lower values in the preterm group than in the term group. All the parameters have increased postnatal in preterm and term newborns; however, this trend was faster and much wider in the term group.

We found significantly lower M indices at birth in preterm than in healthy neonates 24 (*p* = 0.016) or 72 h (*p* < 0.001) after delivery. The M index was also significantly higher in the term group 72 h after delivery than at birth (*p* = 0.001). The M1 indices showed the same trend, but the statistical analyses observed no significant changes (Table 4 and Figure 2).

Like the Myrenne parameters, RBC aggregation properties measured by LORRCA revealed a significantly lower aggregation index (AI) in the preterm group than in term at birth (*p* = 0.012). This significant difference remained positive in the postnatal period for the term group’s benefit (24 h *p* = 0.006; 72 h *p* < 0.001). The AI values in the preterm group stay under the values of the term neonates and the observed increase is also much slower in the preterm group than in the term group. However, we found no significant differences in the other LORRCA parameters (t ½ and gamma value—γ); these showed the same trends as AI (Table 5 and Figure 3).

### 2.4. RBC Deformability

Red blood cell elongation indices (EI) measured at high and middle shear stress were not significantly but relevantly lower at birth in preterm than in term newborns. Perinatal data analyses revealed significantly lower RBC elongation indices in the term group than in preterm neonates 72 h after delivery (30 Pa *p* = 0.011; 16.87 Pa *p* = 0.023; 9.49 Pa *p* = 0.021; 5.33 Pa *p* = 0.046). These differences were significant at the high and middle shear stress range from 30 to 5.33 Pa (Figure 4).

The link between hemorheology in neonates and complications was investigated as well. The predicted probabilities from the combination of 24 h M and EI at 30, 16.87, 9.49, and 5.33 variables were produced by binary logistic regression analysis (*p* = 0.002; degree of freedom = 1). ROC analysis was carried out with the mentioned combined parameters to test the diagnostic power for complications. The analysis indicated AUC with 0.940 (Figure 5A). We have analyzed the same data at 72 h (binary logistic regression showed *p* = 0.001; degree of freedom = 1) and revealed similar results. ROC analysis indicated AUC with 0.939 (Figure 5B).

## 3. Discussion

The exact aetiology for preeclampsia is still unknown, but the evolution of immunology in recent years greatly contributes to the idea of uteroplacental disorders based on oxidative stress, including cytokines (e.g., sFlt-1-soluble fms-like tyrosine-kinase-1, PlGF-placental growth factor) [11,12,13,14]. Due to abnormal trophoblast invasion, smooth muscle tissue remains in the uterine vessel, which results in higher vascular resistance. This abnormality and oxidative stress are in the background of endothelial disturbances and pathological blood flow parameters, which could be measured by ultrasonography and are diagnostic for preeclampsia [15].

Normal perinatal adaptation refers to circulatory and hematological changes in newborns: initiation of ventilation and oxygenation via the lungs, metabolical changes, e.g., the elevation of cortisol production, which is a prerequisite for euglycaemia after delivery, and the development of temperature regulation [16,17]. This transition to extrauterine life requires rapid and complex physiological mechanisms, which must be well organized. Preterm birth causes disturbances in neonatal adaptation and results in longer hospitalisation period after birth because of the numerous complications, i.e., asphyxia, infections, hypoglycaemia, intracranial bleeding, respiratory distress syndrome (RDS), necrotizing enterocolitis (NEC) [18,19], and bronchopulmonary dysplasia (BPD) or retinopathy of prematurity (ROP). Disturbed adaptation mechanisms do not only lead to a complicated perinatal time, but also cause higher cardiovascular risk in adulthood, usually hypertension and diabetes [6,7]. To investigate the different microcirculatory adaptation of our investigational groups, we defined a preterm and a term neonate group. Thereinafter, we discuss the relevant results of the two groups separately.

Term neonates hemorheology during postnatal 72 h.

Several studies in the last decade described the hemorheological status in term neonates’ cord blood samples. First, it was notable that the rheological properties of whole blood in healthy newborns differ markedly from those in adults [20]. In the study of Kuss et al., the RBC aggregation index of term newborns was markedly lower (2.98 ± 2.12) than in adults (14.63 ± 3.50), whereas RBC deformability was better in healthy neonates than in adults. The viscosity parameters determined by Ostwald (WBV) revealed a lower exponent in term neonates (0.94 ± 022) compared to adults (1.01 ± 0.12), as well as the viscosity determined by Newton (PV) was lower in term neonates (1.04 ± 0.16 mPas) than in adults (1.19 ± 0.07 mPas) [21,22]. The findings published by Linderkamp et al. proved that the main difference in rheological properties between neonatal and adult blood lies in the lower plasma viscosity of the neonates resulting in decreased blood viscosity and increased hemoglobin flux through the vessels [23]. Later on, Linderkamp described that specific plasma properties are responsible for the decreased RBC aggregation observed in neonates’ cord blood, while their specific RBC properties (fetal hemoglobin with higher oxygen affinity) do not affect RBC aggregation [24]. Soliman et al. presented their work in 2018. In the publication, the working group described the normal values of plasma viscosity and Myrenne RBC aggregation of healthy neonates’ cord blood [25].

Despite the mentioned studies, we have still no sufficient data about perinatal changes of hemorheological parameters in term infants. The present study aimed to give some hints. Our working group found no significant alterations in term neonates’ hematocrit during the first 72 h of life, although the results refer to polycythaemia. Plasma viscosity data accord with cord blood data of Soliman et al. [25]. Whole blood viscosity showed a continuously increasing tendency without changes in hematocrit, although this difference remained statistically not significant. Higher WBV values could develop by cord clamping delay, which is mostly used at normal vaginal deliveries at term [26]. In our study group, the delayed cord clamping was used in more than 75% of the cases. Both types of RBC aggregation measurement showed an increasing tendency and became actually significant between baseline and 72 h samples in the case of M parameter. Mandelbaum et al. studied 11 term neonates and revealed the same tendency in RBC aggregation, although the increase was not statistically significant on day 5 [27]. In the background, the rising concentration of protein levels (especially fibrinogene) produced by the liver are presumed. We observed no significant differences in RBC deformability measured in term neonates during their first 72 h.

Based on the observed data, macro and microrheological parameters of healthy term newborns do not reach the level of healthy adults within 3 days after birth [20,21,22]. The mentioned findings could strengthen the idea that term neonates have a rather beneficial but continuously shifting microcirculation toward one, which is typical for adults. This healthy microcirculation could easily deal with initial hematological (e.g., polycythaemia) and sweeping cardiopulmonal (blood pressure, initiation of ventilation) changes in the perinatal period.

Preterm neonates hemorheology during postnatal 72 h.

The main goal of our investigation was to describe blood flow properties in macro- and microcirculation of preterm neonates born from preeclamptic pregnancies during the postnatal 72 h. Several studies in the last few decades dealt with hemorheology and preeclampsia, but most of these concentrated on maternal changes. Only a few investigations were published about neonatal effects, and even these describe the hemorheological alterations in cord blood and are controversial.

Previous studies on neonatal hemorheology and preeclampsia presumed that placental insufficiency leads to perinatal hypoxia and limited oxygen supply during the pregnancy stimulating erythropoiesis and raising hematocrit and fibrinogen values. The present study found no significant alterations of preterm neonates’ hematocrit in time, but the values refer to slight polycythaemia. There are published data about the markers of chronic hypoxia (such as polycythaemia, erythropoietin, and nucleated red blood cells), which were elevated in the cord blood of fetuses born to women with preeclampsia [10,28,29,30]. We observed a slight decrease in Htc after 24 h (no significant difference to term newborns), which could be explained by NIC treatment (such as infusions, intravenous antibiotics), but it also highlights the role of delayed cord clamping in the prevention of severe postnatal anaemia requiring transfusion [26,31,32]. WBV showed similar results, supporting the beneficial effect of delayed cord clamping. On the other hand, as a side effect of delayed clamping and immaturity we registered icterus, requiring phototherapy in 76.9% in preterm vs. 5.9% in term cases.

Our investigation on cord blood plasma viscosity showed significantly lower values in preterm neonates than healthy ones, based on impaired protein synthesis capacity of the fetal liver caused by hypoxia and immaturity [33]. Other investigations by Linderkamp et al. indicated that this lower plasma viscosity could not be explained by the neonatal RBCs’ specific membrane and functional properties, which are more often present in immature than term newborns [24,34].

Csorba R. et al. found also lower plasma viscosity and RBC aggregation in stasis and under low shear in the cord blood of the fetus from women with severe preeclampsia compared to the moderate form [33]. The results of the present study support the findings: lower RBC aggregation was revealed at birth compared to other timepoints, although these changes remain statistically not significant. The lower RBC aggregation indices could refer to microcirculatory adaptation in preterm neonates to endothelial dysfunction and higher vascular resistance, which probably helps to maintain blood flow in this preeclamptic disadvantageous milieu. The RBC aggregation values described by Soliman [25] were also significantly lower in preterm than in term neonates’ cord blood. Our results showed a slow but not significant increase in all measured RBC aggregation parameters during the early postnatal 72 h. The observed slow increase in RBC aggregation parameters after delivery could result from the termination of the pathological pregnancy, which stopped the vascular function limiting effect of placental cytokines; however, these values could only near the values and remained under the control group’s results at any time point. The investigated first 72 h period unfolded relevant information about the circulatory adaptation to the extrauterine life, which is slower in preterm babies than in term ones. The results of the RBC deformability measurements showed significant higher EI in the preterm groups at birth compared to term neonates, mainly at high and middle shear stresses, and they remain stable in the first 72 h in preterm neonates. Our results could be in accordance with Linderkamp et al.: preterm infants showed significantly better cord blood RBC deformability compared to term neonates [35]. Thus, both whole cell and membrane deformability appear to be increased in preterm infants, as improved RBC deformability may aid in maintaining adequate blood circulation in preterm infants despite low vascular pressure [30].

Based on our prospective measurements on hemorheological changes in preterm neonates, we could show a presumably pseudo-compensatory macrocirculation (Htc, WBV, PV) and microcirculation (RBC aggregation and deformability). This condition creates a more beneficial milieu for immature circulation to normalize, which is obviously a very slow process. The 24 and 72 h combined data could also predict the occurrence of severe complications, with high specificity and sensitivity. These findings should evoke further attention for the topic and could maybe support the theory that measuring RBC properties during NIC treatment could help to detect when macro and microcirculation of preterm neonates are normalized.

### Strengths and Limitations

The strength of our study is the prospective, case-control design with repeated blood sampling in the early perinatal period to evaluate the kinetics and changes of RBC properties. Using two methods together (Myrenne measuring the increase of light transmission through plasma gaps between RBC aggregates and LORRCA detecting laser backscattering from the RBC aggregates) provides more reliable information on RBC aggregation properties. The low total number of enrolled patients, the investigation on premature neonates, born only to mothers with early-onset preeclampsia, and its single-center nature with local treatment strategies and guidelines could limit the generalizability of our findings. Therefore, the results of the investigation should be considered as data from a pilot study.

## 4. Materials and Methods

### 4.1. Subjects

A total of 13 preterm infants of non-smoking women diagnosed with early-onset preeclampsia treated at the Neonatology Intensive Care Ward at the Department of Obstetrics and Gynaecology, University of Pécs, were involved in this prospective, case-control study. The term group was created by 17 neonates born in healthy age- and gestation-matched pregnancies. Exclusion criteria were any relevant maternal comorbidities (anaemia, chronic hypertension, diabetes mellitus, cardiovascular, hematologic, gynaecologic disorders) and/or fetal abnormalities leading to preterm birth (e.g., twin pregnancy, intrauterine developmental abnormality, intrauterine infection) as well as participation in another study or the lack of signed informed consent from the parents. All the mothers of the newborns participating in our study gave their written informed consent. For further maternal data, see reference [36].

### 4.2. Sample and Data Collection

The sample collection was performed at the same time points in the case of preterm and healthy newborns. To explore the fetal blood rheology, 3 mL umbilical cord blood was obtained within 5 min after removal of the placenta from the uterus into 3 mL EDTA-Vacutainer tubes under standardized conditions. For the postnatal investigations, peripheral blood samples (max 2 mL) were collected by neonatologists 24 ± 3 and 72 ± 3 h after birth, preferably in connection with other routine blood tests (Table 6). We performed hemorheological measurements within 2 h after the blood sampling. Because of neonatologists’ concern about blood sampling volume, we have excluded 6 preterm and 7 term newborns.

At the time of delivery, we registered the neonates’ birth weight and length, Apgar scores at the 1st and 5th minutes, and calculated the weight percentile and verified the diagnosis of IUGR [37,38]. We documented the length of the hospitalization of all neonates after delivery. We recorded mortality and perinatal complications (infections, icterus requiring phototherapy, incidence of ROP, BPD, IVH, and cardiac complications).

### 4.3. Hemorheological Measurements

Hematocrit of the neonates was measured by a micro-hematocrit centrifuge (Haemofuge Heraeus Instr., Hanau, Germany) [39].

Whole blood viscosity (WBV) and plasma viscosity were determined by the Brookfield DV-III Ultra LV (Brookfield Engineering Labs, Middleboro, MA, USA) programmable rotational viscometer at 37 °C. Plasma viscosity was measured only at birth in both groups because of the large amount of required blood [39].

Red blood cell aggregation was measured with Myrenne (MA-1 Aggregometer, Myrenne GmbH, Roentgen, Germany) and LORRCA (Laser-assisted Optical Rotational Cell Analyzer; R&R Mechatronics, Hoorn, The Netherlands) aggregometers using blood samples with native hematocrit. Myrenne measures the infrared light transmitted through the plasma gaps between RBC aggregates and provides two dimensionless indices at room temperature (M, aggregation at stasis, and M1, aggregation at low shear) using the Schmid-Schönbein principle; both M and M1 are increased with enhanced red blood cell aggregation. The measurement required 30 μL whole blood [39,40]. LORRCA aggregometer needed 1 mL whole blood and determined RBC aggregation index (AI) at 37 °C via syllectometry (i.e., laser backscattering versus time). The RBC disaggregation threshold (γ) describes the minimal shear rate needed to prevent RBC aggregation formation [39,41].

Erythrocyte deformability was characterized with LORRCA ektacytometer at 37 °C. Moreover, a 20 μL blood sample was diluted in a viscous medium (polyvinylpyrrolidone) and injected between the cylinders of the instrument. The provided nine values of elongation indices (EI) at the shear stress ranging from 0.3 to 30 Pa refer to the RBC diffraction pattern changing from circular to elliptical shape [39,42].

### 4.4. Statistics

A sample size and power analysis was performed for term newborns and for neonates born from preeclamptic pregnancy using PS program version 3.1.2. For the sample size of *n* = 17 patients, it needed to detect a true difference of d = 10.33 in LORCA AI with 80.61% power, where type I error probability was α = 0.05. For the sample size of *n* = 13 patients, it needed to detect a true difference of d = 10.42 in LORCA AI with 85.08% power, where type I error probability was α = 0.05.

Statistical analysis was evaluated by IBM SPSS Statistics^®^ 27.0. Continuous variables are reported as mean ± standard deviation (SD), with categorical variables as frequencies and percentages. After testing the normality by the Kolmogorov–Smirnov test, a one-way repeated ANOVA statistical test (Tamhane post-hoc test) and chi-square test were used to compare differences between groups. *p* ≤ 0.05 was considered statistically significant.

## 5. Conclusions

Disturbed neonatal adaptation is the main complication of preeclamptic pregnancies. Our results described the changes in hemorheological parameters in preterm and term infants, which could refer to slow and disturbed circulatory adaptation in preterm neonates. We generally validate the findings of previous studies on the topic, but we offer new details about the process of neonatal circulatory adaptation in the early perinatal period.

## Figures and Tables

**Figure 1 ijms-24-08496-f001:**
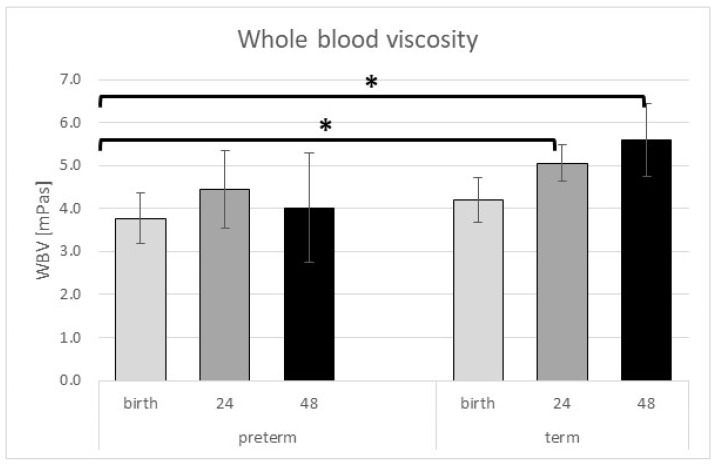
Whole blood viscosity showed an increase in the term group postnatal. * represents a significant *p* value (*p* < 0.05). Data are shown as mean ± standard deviation.

**Figure 2 ijms-24-08496-f002:**
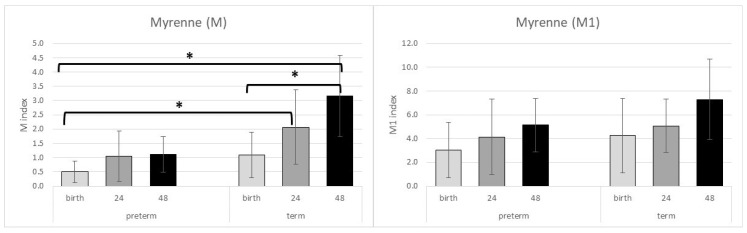
Myrenne indices showed an increase with time; however, in preterm neonates, this trend is slower than in the term group. * represents a significant *p* value (*p* < 0.05). Data are shown as mean ± standard deviation.

**Figure 3 ijms-24-08496-f003:**
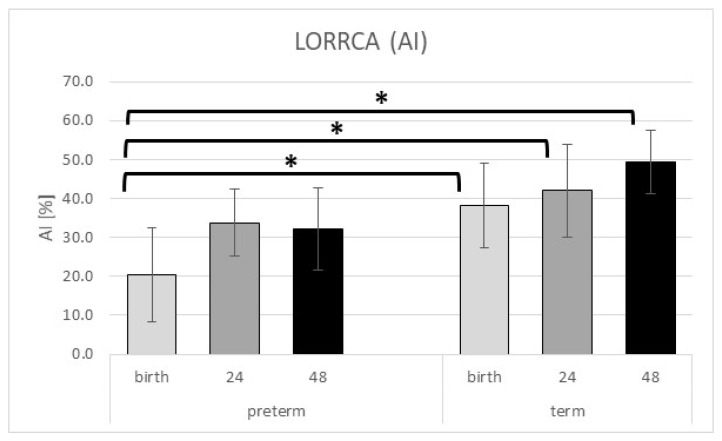
LORRCA AI significantly increased in the postnatal period. * represents a significant *p* value (*p* < 0.05). Data are shown as mean ± standard deviation.

**Figure 4 ijms-24-08496-f004:**
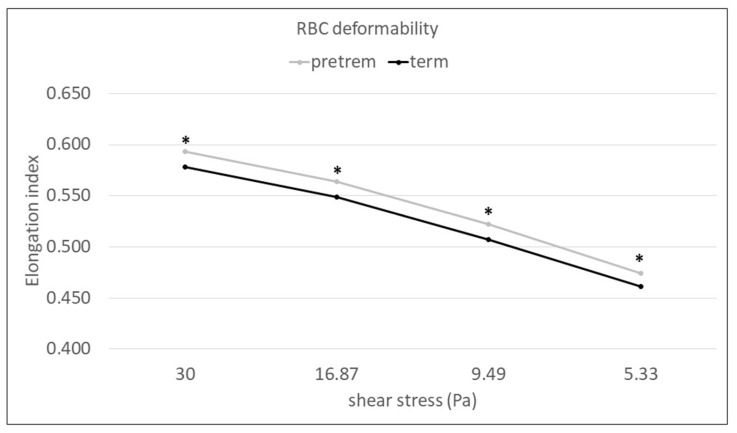
LORRCA EI in the preterm group measured at 72 h after delivery and term neonates at birth. * represents significant *p* value (*p* < 0.05). Data are shown as mean ± standard deviation (df = 5.71 at all shear stress) (overall *p* value EI30 and EI16.87 < 0.001; EI9.49 and EI5.33 = 0.001; EI3 = 0.002; EI1.69 = 0.007; EI0.95 = 0.008; EI0.53 = 0.118; EI0.3 = 0.385).

**Figure 5 ijms-24-08496-f005:**
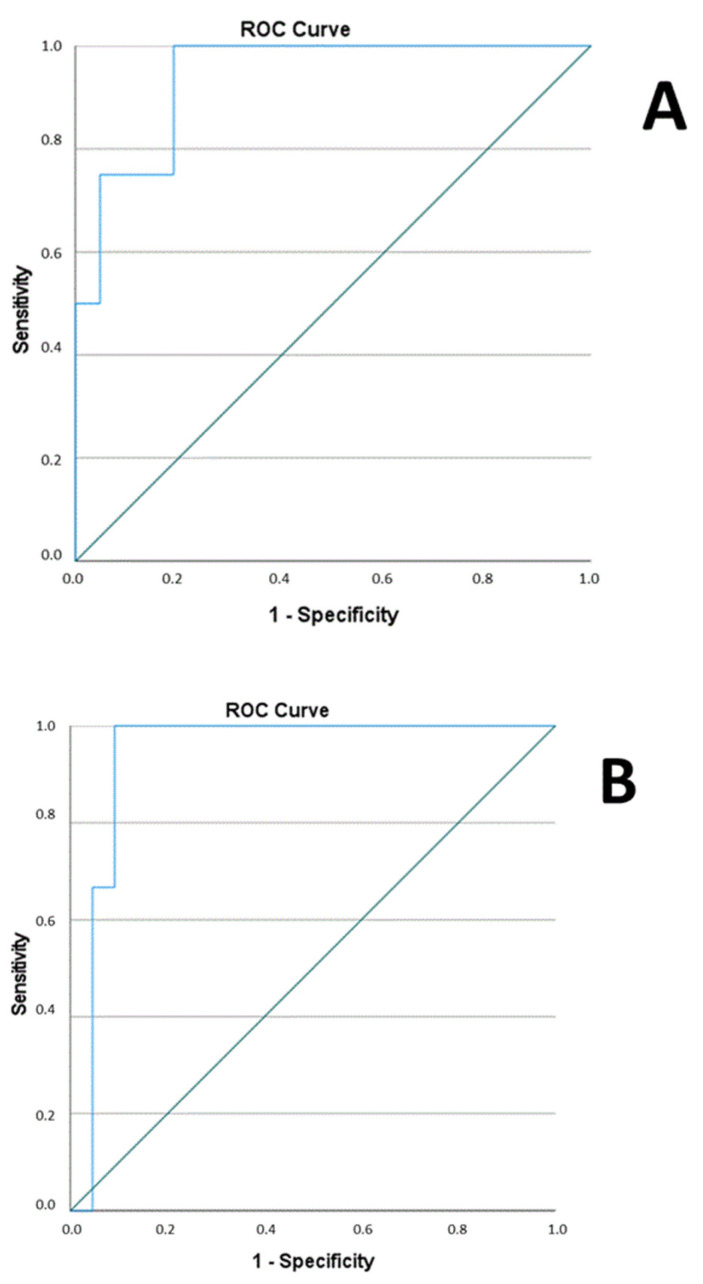
(**A**). ROC curve for combined hemorheological parameters at 24 h and complications to express predicted probability. (**B**). ROC curve for combined hemorheological parameters at 72 h and complications to express predicted probability.

**Table 1 ijms-24-08496-t001:** Statistical analyses (in case of continuous variables ANOVA, in case of categorical variables chi-square test) of the neonatal demographical data at birth. Parameters are shown as mean ± SD and percentage.

Parameter	Preterm Neonates (*n* = 13)	Term Neonates (*n* = 17)	Degree of Freedom	*p* Value
gestational age at birth (weeks)	30.23 ± 3.09	39.06 ± 1.14	5.84	**<0.001**
Caesarean section	100%	35.30%	1	**<0.001**
delayed cord clamping	76.92%	76.47%	1	0.977
birth weight (grams)	1306.92 ± 551.94	3420.00 ± 367.13	1.28	**<0.001**
birth length (cm)	36.67 ± 5.69	49.77 ± 3.31	1.27	**<0.001**
1 min Apgar	7.0 ± 1.29	8.82 ± 0.64	1.28	**<0.001**
5 min Apgar	8.69 ± 0.63	9.88 ± 0.33	1.28	**<0.001**
length of hospitalization (day)	45.15 ± 26.08	3.35 ± 0.61	1.28	**<0.001**
weight percentile at birth (%)	10.73 ± 1.95	52.49 ± 20.85	1.28	**<0.001**
incidence of IUGR (%)	69.20	0.00	5	**<0.001**
infection after birth	38.50%	0.00%	1	**0.005**
icterus treated by phototherapy	5.90%	76.9%	1	**<0.001**
incidence of ROP, BPD, IVH, cardiac and gastrointestinal complications	53.85%	0.00%	1	**<0.003**

**Table 2 ijms-24-08496-t002:** Statistical analyses of the neonatal hematocrit data at birth, 24 and 72 h postnatal. Parameters are shown as mean ± SD.

	Time	Hematocrit [%]	Degree of Freedom	*p* Value
preterm neonates	at birth	49.91 ± 5.13	5.71	0.416
	24 h postnatal	50.17 ± 8.53		
	72 h postnatal	45.89 ± 7.71		
term neonates	at birth	50.50 ± 4.12		
	24 h postnatal	50.08 ± 3.07		
	72 h postnatal	50.44 ± 4.16		

**Table 3 ijms-24-08496-t003:** Statistical analyses of the neonatal whole blood and plasma viscosity data at birth, 24 and 72 h postnatal. Parameters are shown as mean ± SD. NA = not applicable, dF = degree of freedom.

	Time	WBV [mPas] (dF = 5.39) (Overall *p* Value < 0.001)	PV [mPas] (dF = 3.12) (Overall *p* Value = 0.041)
preterm neonates	at birth	3.77 ± 0.59	0.91 ± 0.04
	24 h postnatal	4.45 ± 0.90	NA
	72 h postnatal	4.01 ± 1.27	NA
term neonates	at birth	4.21 ± 0.52	0.95 ± 0.04
	24 h postnatal	5.05 ± 0.43	NA
	72 h postnatal	5.6 ± 0.84	NA

**Table 4 ijms-24-08496-t004:** Statistical analyses of the neonatal M indices at birth, 24 and 72 h postnatal. Parameters are shown as mean ± SD, dF = degree of freedom.

	Time	M Index (dF = 5.71) (Overall *p* Value < 0.001)	M1 Index (dF = 5.71) (Overall *p* Value = 0.007)
preterm neonates	at birth	0.49 ± 0.59	3.04 ± 2.33
	24 h postnatal	1.05 ± 0.88	4.13 ± 3.19
	72 h postnatal	1.11 ± 0.62	5.14 ± 2.26
term neonates	at birth	1.09 ± 0.80	4.25 ± 3.13
	24 h postnatal	2.07 ± 1.30	5.06 ± 2.25
	72 h postnatal	3.17 ± 1.42	7.30 ± 3.38

**Table 5 ijms-24-08496-t005:** Statistical analyses of the neonatal LORRCA aggregation index, t ½, and gamma value at birth, 24 and 72 h postnatal. Parameters are shown as mean ± SD, dF = degree of freedom.

	Time	AI (dF = 5.61) (Overall *p* Value < 0.001)	t ½ (dF = 5.61) (Overall *p* Value < 0.001)	γ (dF = 5.61) (Overall *p* Value = 0.002)
preterm neonates	at birth	20.38 ± 11.97	19.83 ± 11.81	41.59 ± 2.57
	24 h postnatal	33.74 ± 8.61	9.70 ± 4.86	48.25 ± 13.65
	72 h postnatal	32.19 ± 10.69	9.50 ± 4.13	42.08 ± 2.46
term neonates	at birth	38.19 ± 10.88	7.39 ± 3.32	45.16 ± 6.49
	24 h postnatal	42.10 ± 11.92	6.66 ± 3.93	51.36 ± 12.72
	72 h postnatal	49.45 ± 8.20	4.31 ± 1.58	80.77 ± 53.26

**Table 6 ijms-24-08496-t006:** Blood sampling time points. A total of 13 infants of women diagnosed with early-onset preeclampsia and 17 infants of the gestational age-matched healthy non-smoking pregnant women.

	Blood Sampling		
13 preterm neonates	at birth (umbilical cord blood)	24 ± 3 h after delivery (venous)	72 ± 3 h after delivery (venous)
17 healthy neonates	at birth (umbilical cord blood)	24 ± 3 h after delivery (venous)	72 ± 3 h after delivery (venous)

## Data Availability

The authors can provide the dataset and analyses generated during the study upon request.

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
