# Peer review of "The Influence of Early Onset Preeclampsia on Perinatal Red Blood Cell Characteristics of Neonates"

_ijms, 2023, doi:10.3390/ijms24108496_

Round 1

Reviewer 1 Report

The methods remain confusing and despite edits, it is still unclear.

In this study authors evaluated hemorheological parameters in newborns of mothers with early-onset preeclampsia by investigating red blood cell (RBC) properties. This is an interested topic but I have some concerns regarding the study methods.

The methods remain confusing and despite edits, it is still unclear.

I

1.       As the authors mentioned their main goal was to describe the changes of hemorheological factors especially RBC properties in preterm neonates born from early-onset preeclamptic mothers during a short perinatal period. Given the fact that the hemorheological parameters of premature infants differ from those of full-term infants why they choose as a control term neonates and not preterms born to mothers without preeclampsia?. This fact raises questions about the selection bias of the study. Is there performed any adjustment for gestational age?.

Author Response

Answer to reviewer 1 per report ijms-2271334.

Title:

The influence of early onset preeclampsia on perinatal red blood cell characteristics of neonates

Authors:

Barbara Sandor, Beata Csiszar, Gergely Galos, Simone Funke, Dora Kinga Kevey, Matyas Meggyes, Laszlo Szereday, Kalman Toth

Dear Reviewer!

We would like to thank you for the support and valuable critics in your previous and present review. We highly appreciate your work, which was fundamental for the new manuscript.

We would like to answer to your reflections.

The methods remain confusing and despite edits, it is still unclear.

In this study authors evaluated hemorheological parameters in newborns of mothers with early-onset preeclampsia by investigating red blood cell (RBC) properties. This is an interested topic but I have some concerns regarding the study methods.

The methods remain confusing and despite edits, it is still unclear.

I

  1. As the authors mentioned their main goal was to describe the changes of hemorheological factors especially RBC properties in preterm neonates born from early-onset preeclamptic mothers during a short perinatal period. Given the fact that the hemorheological parameters of premature infants differ from those of full-term infants why they choose as a control term neonates and not preterms born to mothers without preeclampsia?. This fact raises questions about the selection bias of the study. Is there performed any adjustment for gestational age?.

As we have described in our previous answer for You the main goal was to describe perinatal hemorheological changes in term and preterm neonates.

We agree, that term and preterm neonates should not be compared to each other, therefore we have removed the confusing word “control” from the manuscript and used the word “term” instead. Maybe this change (and the separated discussion of the two groups) could indicate more, that the main goal was not to describe the univocally differences between healthy and preterm (for any reason) neonates, but the perinatal changes in both groups. Thank You for pointing this out.

As we have mentioned it before it was not intended in the present study to compare preeclamptic preterm neonate’s data to other preterm newborns, but obviously it can be an interesting topic.

The gestational match was only performed in case of the mothers.

We hope that our answers and corrections fulfill your requirements.

Sincerely yours,

            On behalf of the authors Barbara Sándor

Reviewer 2 Report

Comments and Suggestions for Authors

The manuscript is an original article that addresses the assessment of red blood cell characteristics in newborns of early-onset preeclamptic mothers. I am glad that the authors made essential improvements to the manuscript after resubmission. They addressed the critical points that I mentioned previously.

In this sense, they performed a sample size and power analysis, and the dF=degree of freedom was introduced in all 5 tables. Moreover, two ROC curves were inserted in the manuscript to express the predicted probability.

Also, significant portions of the discussion chapter were rewritten, thus bringing a better approach to the results obtained.

However, I recommend inserting in the chapter „Strengths and limitations”  that the study is a pilot one.

In conclusion, the changes made to the article are significant and can support its publication in the journal.

Moderate editing of the English language is needed.

Author Response

Answer to reviewer 2 per report ijms-2271334.

Title:

The influence of early onset preeclampsia on perinatal red blood cell characteristics of neonates

Authors:

Barbara Sandor, Beata Csiszar, Gergely Galos, Simone Funke, Dora Kinga Kevey, Matyas Meggyes, Laszlo Szereday, Kalman Toth

Dear Reviewer!

We would like to thank you for the support and valuable critics in your previous and present review. We highly appreciate your work, which was fundamental for the new manuscript.

We would like to answer to your reflections.

The manuscript is an original article that addresses the assessment of red blood cell characteristics in newborns of early-onset preeclamptic mothers. I am glad that the authors made essential improvements to the manuscript after resubmission. They addressed the critical points that I mentioned previously.

In this sense, they performed a sample size and power analysis, and the dF=degree of freedom was introduced in all 5 tables. Moreover, two ROC curves were inserted in the manuscript to express the predicted probability.

Also, significant portions of the discussion chapter were rewritten, thus bringing a better approach to the results obtained.

However, I recommend inserting in the chapter „Strengths and limitations”  that the study is a pilot one.

In conclusion, the changes made to the article are significant and can support its publication in the journal.

Thank you again for the approval of our new methods, statistics and discussion part. We have completed the “Strengthen and limitations” part with a sentence, referring to the pilot style of the study (marked red).

Round 2

Reviewer 1 Report

As the authors mentioned the aim of the study was to describe perinatal hemorheological changes in term and preterm neonates, so I believe that the title of the study should be modified to reflect its purpose and moreover the fact that the study includes only premature neonates born to the mothers with preeclampsia, should be defined as a study limitation.

Author Response

Answer to reviewer 1 per report ijms-2271334.

Title: The influence of early onset preeclampsia on perinatal red blood cell characteristics of neonates

Authors: Barbara Sandor, Beata Csiszar, Gergely Galos, Simone Funke, Dora Kinga Kevey, Matyas Meggyes, Laszlo Szereday, Kalman Toth

 Dear Reviewer!

We would like to thank you for the suggestions You have made. We appreciate your efforts and time-to-spent for this review.

We would like to answer to your reflections.

1. As the authors mentioned the aim of the study was to describe perinatal hemorheological changes in term and preterm neonates, so I believe that the title of the study should be modified to reflect its purpose and moreover the fact that the study includes only premature neonates born to the mothers with preeclampsia, should be defined as a study limitation.

We have updated the title of the manuscript, hopefully it describes the topic and aims better now.

Short-term perinatal changes of red blood cell characteristics in term, and preterm neonates due to early-onset preeclampsia.

We have also rewritten the "Strengths and limitations" part, and completed it with the fact, that only preeclamptic preterm newborns were investigated.

We hope that our answers and corrections fulfill your requirements.

Sincerely yours,

            On behalf of the authors Barbara Sándor